# Leadership and Turnover Intentions in a Public Hospital: The Mediating Effect of Organisational Commitment and Moderating Effect by Activity Department

**Patrícia Martins [1], Generosa Nascimento [2] and Ana Moreira [3,\*]**

[1] ISCTE-Instituto Universitário de Lisboa, Avenida das Forças Armadas, 1649-026 Lisbon, Portugal
[2] Business Research Unit, ISCTE-Instituto Universitário de Lisboa, Avenida das Forças Armadas, 1649-026 Lisbon, Portugal
[3] School of Psychology, ISPA—Instituto Universitário, Rua do Jardim do Tabaco 34, 1149-041 Lisbon, Portugal
\* Correspondence: amoreira@ispa.pt

**Abstract:** This research aimed to study the effect of leadership (transformational and transactional) on turnover intentions and whether this relationship is mediated by organisational commitment and moderated by the department of activity. To this end, it was hypothesized that: (1) leadership has a negative and significant association with turnover intentions; (2) leadership has a positive and significant association with affective organisational commitment; (3) organisational commitment has a negative and significant association with turnover intentions; (4) organisational commitment has a mediating effect on the relationship between leadership and turnover intentions; (5) the department to which the employee belongs has a moderating effect on the relationship between leadership and organisational turnover intentions. The sample consists of 477 participants working at the Hospital Professor Doutor Fernando Fonseca (HFF) in Portugal, performing their functions in several departments. This is a quantitative, correlational, and cross-sectional study. The results indicate that transformational leadership has a negative and significant effect on turnover intentions and a positive and significant effect on affective and normative commitment. Transactional leadership negatively and significantly affects all three components of organisational commitment. Affective and normative commitment has a negative and significant effect on turnover intentions. Calculative commitment has a positive and significant effect on turnover intentions. Affective commitment and calculative commitment have a partial mediating effect on the relationship between transformational leadership and organisational turnover intentions. The department to which the employee belongs does not have a moderating effect on the relationship between leadership and turnover intentions. It has been proven that leadership plays an extremely important role in the smooth running of this hospital unit.

**Keywords:** leadership; organisational commitment; turnover intentions; department of activity

## 1. Introduction

Healthcare organisations provide complex and highly specialized services and encounter vulnerable and dependent users. Thus, users are the raison d'être of the organisation, and all interventions are designed to meet their needs increasingly effectively. These are changing times, and healthcare organisations are increasingly seen as businesses striving daily for prestige and striving to be competitive in the job market. The quality of services provided by healthcare organisations is becoming increasingly important as more users now have more access to healthcare services than ever before, with more choices and greater demands. Providing high-quality services at the lowest possible cost is the logistical capability of all organisations. The production process of health services has certain characteristics that constitute and determine the importance of human resources in this specific labour market; therefore, the existence of adequate, competent, and motivated human resources cannot be ignored, as it is crucial to the performance of the organisation

(Caetano and Vala 2002). This new approach implies a review of social and economic infrastructures, employment practices, remuneration systems, and employee training. However, high financial costs and limited human resources pose several challenges to healthcare organisations, especially in the nursing and healthcare sectors.

In the last two years, the crisis caused by the COVID-19 pandemic has affected all organisations and employees, especially in the healthcare sector. This situation has caused many employees to rethink the continuity in their work organisation. This situation exacerbates organisations' serious problems today: high turnover and the consequent loss of experienced staff, representing high replacement costs (Reiche 2008). High turnover rates in healthcare can have a negative impact on meeting the needs of users as well as on the quality of care. Staff turnover has financial consequences for the development of organisational activities, mainly in the form of costs, reduced productivity, and poor quality of service.

Successful leadership in organisations is becoming increasingly challenging. This is due to the historical context in which we find ourselves. Speed and adaptability have marked healthcare professionals in an uncertain and constantly changing environment.

In turn, effective leadership can influence the most valuable outcomes of the organisation, reducing employee turnover intentions and increasing customer satisfaction and organisational effectiveness (Asrar-ul-Haq and Kuchinke 2016). In effective leadership, the leader uses his knowledge and experiences to contribute to the growth of the organisation and the employees (Perez and Oliveira 2015). An effective leader influences without exercising authority over the collaborators. When he influences, he inspires confidence and motivates his followers to do what he wants them to do because they want to do it. An effective leader leads employees so that they reach goals they never imagined they could achieve (Perez and Oliveira 2015). The leadership style adopted in an organisation is considered one of the organisational variables that most enhances the motivation of employees, thus increasing their affective organisational commitment, becoming a success factor for the organisation to achieve the desired results (Fiaz et al. 2017). According to Meyer et al. (2002), one of the consequences of affective commitment is a decrease in turnover intentions because when the employee feels emotionally attached to the organisation where he or she works, he or she feels that he or she should stay there, which will lead to a decrease in intentions to leave, as well as in unjustified absences from work. For Meyer et al. (2004), of the three components of organisational commitment, the one that is most related to all the results is affective commitment.

Responsible leadership contributes to employees' affective commitment, increasing organisational productivity (Haque et al. 2017) and decreasing turnover intentions (Mercurio 2015). Due to their differentiating aspects, healthcare organisations are a real challenge for any leader. HFF was chosen because it is in a peripheral area of Lisbon, the capital of Portugal. This hospital is in one of the country's largest and most diverse population centres. This human and territorial range allows us to live a set of experiences in a hospital environment that has enriched everyone. In addition, at this moment and after the COVID-19 pandemic, the hospital sector in Portugal is going through a great crisis, with many professionals (especially doctors and nurses) leaving public hospitals. What leads us to try to understand if the department moderates the relationship between leadership and turnover intentions is that some departments are confronted with a great overload of service due to a lack of professionals. The department where this problem has been felt most frequently in the Lisbon and Tagus Valley region is the Women and Children's department. The problem is mirrored in several regional hospitals since they work in a network.

The main objective of this research is to study the effect of leadership style (transformational and transactional) on employees' turnover intentions, as well as whether organisational commitment is the mechanism that explains this relationship. Another objective is to test whether the department to which the employee belongs has a moderating effect on these relationships. With this study, we intend to give those who run this hospital

some guidelines that will contribute to the progressive improvement of the valorisation of the professionals who, every day, without exception, dedicate their lives to helping others.

## 2. Theoretical Framework and Research Hypotheses

### 2.1. Health Services Organisations

According to the Ministry of Health ([MS], 1998), a hospital is an organisation that has the primary purpose of providing healthcare, operates effectively and in a well-defined way with other institutions in the healthcare delivery network, and aims to provide a wide range of comprehensive services 24 h a day. It consists of technical and human resources and develops its activities through diagnosis, treatment and rehabilitation in inpatient or outpatient settings; Morgado et al. (2009) consider these instruments essential components in the healthcare system because they consume financial resources. They play an important role in education and training in the health sector and in strengthening research (Morgado et al. 2009). Healthcare organisations are complex structures because of their missions and expectations and the highly autonomous, multidisciplinary teams that use advanced technology to provide patients with preventive, therapeutic, and rehabilitative healthcare services. Health systems include healthcare, disease prevention, health promotion, and efforts to influence other sectors that may affect health. Health system characteristics can contribute to knowledge transfer and transformation, comparisons, and added value in health policy and strategy (Hassenteufel and Palier 2007; Odier 2010).

Portuguese public hospitals evolved, as seen in first-world countries, in terms of differentiation of facilities, provision of technology and human resources. However, organisational, management and operational models remained entangled in a web of bureaucracy, self-reproducing command and control within the organisation, and persistent centralism in health management.

### 2.2. Leadership

Leading is how an individual influences others to achieve a common goal for a group or organisation (Northouse 2013). The essence of leadership is influencing followers. The role of leadership power is to act as a mechanism of influence (Bass and Bass 2008). Leadership is the process in which one person influences others to achieve common goals, without relationships of force but of power/influence in the sense of the ability to change the attitudes, behaviours, and beliefs of others (Northouse 2013). From Avolio et al. (2009) perspective, leadership is fundamental to the functioning of individuals, teams, and organisations, yet leadership style is essential at all organisational levels (micro, meso, and macro).

According to Nascimento (2021), leadership should not be an attribute, but an intentional and constructive process designed to improve organisational (or functional) outcomes. Leaders must maximize the organisation's effectiveness in the present while providing the resources and capabilities that will distinguish it in the future. Such sustainability requires promoting efficiency, reliability, innovation, adaptation to the external environment and a continuous assessment of human capital (people). Furthermore, according to Nascimento (2021), the management of healthcare organisations has specific characteristics inherent to the characteristics of these organisations. They aim to provide users with health services for prevention, treatment, and rehabilitation using advanced technologies. They are also a space for teaching, learning, research, and innovation. The complexity of their mission and structure, the high degree of autonomy and their teams' technical and scientific competence require leadership with deep knowledge and experience in defining policies and strategies and managing people and processes.

### 2.2.1. Transformational Leadership

Transformational leadership is characterized as a leadership procedure that leads to changes in attitudes and behaviours practised by employees in the organisations where it is practised, according to Burns (1978). Judge and Piccolo (2004) consider that, for many authors, a more effective leadership.

The perception of Transformational Leadership is positively related to the practice of rational influencing behaviours (Epitropaki and Martin 2013). Transformational leadership can be seen as the most important factor in explaining how leaders can lead their employees to perform desirable behaviour and achieve optimal levels of performance (Moynihan et al. 2012). Transformational leaders use rational techniques, allowing their employees to participate in decision-making in the work context whenever possible.

Transformational leadership is characterised by positive, proactive leadership. It consists of four related behaviours: idealising behaviours, idealising influence, inspiring motivation, intellectual stimulation, and individualised consideration, which, when well-articulated, correlate with outcomes indicating follower, team, and organisational effectiveness (Tepper et al. 2018). These behaviours encourage ethical behaviour, increase employee motivation and optimism, encourage independent thinking, and focus on everyone's needs (Robertson 2018). As a result, transformational leaders demonstrate a moral foundation in their words and actions. They are viewed by their subordinates as role models, motivating them to go above and beyond for the organisation's good (Lin et al. 2017).

### 2.2.2. Transactional Leadership

Along with transformational leadership, Burns (1978) introduced the concept of transactional leadership (Judge and Piccolo 2004). Bass (1997) refers to this leadership as a process of extrinsic exchanges and rewards between leaders and followers, through which the leader clarifies the expectations and goals to be achieved and the rewards in their pursuit. According to the literature, the main difference between the two concepts is the exchange relationship between leader and follower. Differing from transactional leadership, transformational leadership motivates followers' efforts not through material rewards but through higher values such as praise, trust, loyalty, respect, and commitment (e.g., Bass 1985; Bass and Avolio 1993; Podsakoff et al. 1986, 1996).

The relationship between leader and followers becomes transactional when the leader controls rewards and contingencies (Moriano et al. 2014). According to Bass and Bass (2008), transactional leadership comprises two components: contingent reward and active management by exception. Contingent reward tells us the extent to which the leader correctly recognizes, and rewards good work. Active management by exception refers to the leader's behaviours to anticipate and resolve errors and failures.

Transactional leaders bypass innovations and breakthroughs to keep their led focused on performing their tasks (Parker 2011). The leader extrinsically motivates his or her followers, which can lead to creativity and innovation remaining at minimal levels (Jung 2001). For Moriano et al. (2014), a leader with a transactional leadership style ensures consistency with their status rather than with innovation or change.

### 2.3. Turnover Intentions

As for turnover intentions, these are considered the best predictor of voluntary organisational departure, leading to a negative impact on organisational performance, which motivates great interest in studying this construct (Long et al. 2012; Park and Shaw 2013). Turnover intentions are understood as employees' willingness to leave the organisation where they are and start looking for a new workplace (Tett and Meyer 1993; Benson 2006).

Organisational characteristics such as perceived organisational support (Hui et al. 2007), organisational culture (Islam et al. 2012), perceived human resource management practices (Heavey et al. 2013), organisational commitment, job satisfaction, autonomy, and responsibility (Shore and Martin 1989) are antecedents of turnover intentions.

The goal of any organisation should be to increase employee commitment and avoid the constraints of turnover. However, any company, regardless of its size, goes through situations of dismissal, whether on the employee's initiative, for various reasons of personal or professional nature or when an organisation no longer needs the services of that employee. Leaving can lead to decreased service quality since the new employee needs time

to fully understand all the processes, which interferes with colleagues and job performance. When an employee leaves the organisation, it is not only the employee himself lost but also his specific skills and knowledge about his duties (Iqbal et al. 2017). In addition, new recruitment and selection processes are associated with high human and financial costs. A high turnover rate can cause instability and uncertainty regarding the working conditions offered by the organisation, both internally and externally (Indrasari et al. 2019).

Leadership and Turnover Intentions

Several studies have suggested that leadership, especially transformational leadership, has advantages in reducing intentions to leave the organisation, as it leads to followers wanting to stay with the organisation. For Silva (2001) and Mobley (1992), the leader can be an important source or enabling factor concerning the goals and rewards achieved by the employees, playing an important role in reducing their turnover intentions. The transformational leader encourages his employee to take a critical role in decision-making (Özaralli 2003), which raises their job satisfaction levels (Kim 2002). The relationship between leadership and turnover intentions can be interpreted based on the Resource Conservation Theory. This theory argues that employees seek to create, protect, and maintain their resources to protect themselves or replace losses, as well as to acquire new resources because it is a fact of having these resources that will lead them to obtain positive results (Hobfoll 1989; Pinto and Chambel 2008). Let us consider that leadership can be considered a resource. It is logical that if an employee perceives the leader's leadership style as correct, he or she does not want to lose this resource, which leads to a decrease in turnover intentions.

According to Park and Pierce (2020), transformational leadership negatively and significantly affects turnover intentions. Moreover, Yücel (2021) proved that transactional leadership is a reducer of organisational exit intentions. Koesmono (2017) proved that transactional leadership positively and significantly affects turnover intentions. Following the reasoning of these authors, the following hypotheses were formulated:

**Hypothesis 1.** *Leadership style has a significant effect on turnover intentions.*

**Hypothesis 1a.** *Transformational leadership has a negative and significant effect on turnover intentions.*

**Hypothesis 1b.** *Transactional leadership has a positive and significant effect on turnover intentions.*

*2.4. Organisational Commitment*

Several definitions of organisational commitment reflect three main themes: attachment to the organisation, the perceived value of leaving the organisation, and the commitment to staying with the organisation (Meyer and Allen 1991).

From Meyer and Allen's (1991) perspective, the conceptualisation of organisational commitment is based on three components: affective, calculative, and normative. These three approaches have in common that they view commitment as a psychological state that characterizes the relationship between employees and the organisation and influences the decision of whether to stay with the organisation (Meyer and Allen 1991).

Affective organisational commitment is interpreted as an employee's emotional connection, identification, and involvement with the organisation in which they work (Meyer 2014). This type of organisational commitment is driven by positive emotions about the organisation, resulting from communication between the organisation and its employees (Colquitt et al. 2014) regarding how an organisation treats them. When employees feel part of the organisation's "family," creating personal meaning and belonging, they do the work they do and thus contribute to the organisation's success (Meyer et al. 2004). Affective commitment is developed through previous work experiences in the organisation,

experiences that satisfy employees' psychological needs, making them feel comfortable within the organisation and competent in performing their jobs (Ng 2015).

Calculative commitment refers to the concept of the cost of leaving the organisation. Thus, employees whose primary relationship with the organisation is calculative commitment stay because they need to. This component of organisational commitment develops through a series of investments that may or may not be directly related to employment (Meyer and Allen 1991). They may include investments in specific skills that cannot be transferred to other organisations, the loss of attractive benefits, and the loss of retirement privileges, which may be perceived as a potential cost of leaving the organisation (Meyer and Allen 1991).

Normative commitment is associated with a sense of obligation to stay in the organisation. Employees with high levels of normative commitment tend to stay in the organisation because they feel they have this duty (Meyer and Allen 1991). The socialisation experiences that lead to the sense of obligation may begin with observing role models in the organisation or socialisation experiences present in the family or culture (Meyer et al. 2007). Employees who have been led to believe, via various prior organisational practices, that the organisation deserves loyalty will be more willing to develop a solid normative commitment (Meyer et al. 2002). Meyer and Parfyonova (2010) propose that Normative Commitment constitutes a moral duty and obligation to "pay the debt."

However, these three forms of commitment need not necessarily manifest themselves in isolation, and all three can coincide with varying degrees of importance.

### 2.4.1. Leadership and Organisational Commitment

Transformational leadership can lead to followers feeling high levels of organisational commitment and job satisfaction. Transformational leaders inspire followers to transcend their interests and orient themselves towards collective interests, which leads them to develop high levels of affective commitment to the organisation where they work (Bass 1985). According to Fiaz et al. (2017), effective leadership leads to employees feeling a higher level of affective commitment.

Several studies suggest that effective leadership leads to increased affective commitment levels. They have found a positive relationship between various leadership approaches and subordinates' attitudes (such as organisational commitment), motivation, and performance (Haque et al. 2017). A values-based leadership style, such as transformational and ethical leadership, has been shown to have a positive and significant association with affective and normative organisational commitment (Hashim et al. 2017). However, Zaraket and Sawma (2018) did not find a significant association between transformational leadership and calculative commitment. The relationship between transformational leadership and organisational commitment (affective and normative) can be interpreted based on the premise of social exchanges (Blau 1964) and the norm of reciprocity (Gouldner 1960). When the employee perceives their leader's leadership style as transformational, their way of rewarding it will be to stay in the organisation.

From the perspective of Ngunia et al. (2006), transactional leadership, based on extrinsic exchanges and rewards between leaders and followers, has a significant and negative effect on the three components of organisational commitment.

The following hypotheses were thus formulated:

**Hypothesis 2.** *Leadership style has a significant effect on organisational commitment.*

**Hypothesis 2a.** *Transformational leadership has a positive and significant effect on affective commitment.*

**Hypothesis 2b.** *Transactional leadership has a negative and significant effect on affective commitment.*

**Hypothesis 2c.** *Transformational leadership has a positive and significant effect on calculative commitment.*

**Hypothesis 2d.** *Transactional leadership has a negative and significant effect on calculative commitment.*

**Hypothesis 2e.** *Transformational leadership has a positive and significant effect on normative commitment.*

**Hypothesis 2f.** *Transactional leadership has a negative and significant effect on normative commitment.*

2.4.2. Organisational Commitment and Turnover Intentions

According to Meyer and Allen's (1991) three-dimensional model, one of the consequences of organisational commitment is reduced turnover intentions. When an employee feels connected to the organisation, they feel that they must stay in the organisation, leading to lower turnover intentions and unjustified absences (Meyer et al. 2002).

Several authors believe that the main interest in studying organisational commitment is its impact on turnover intentions, given that employees who are committed to the organisation want to stay there (Moreira et al. 2020). In a study by Wasti (2003), the results indicated that although all three components of organisational commitment have a negative effect on turnover intentions, affective organisational commitment has the most substantial relationship. Moreover, Moreira and Cesário (2021) concluded that affective organisational commitment is the best reducer of turnover intentions among the three components of organisational commitment. The following hypothesis is thus formulated:

**Hypothesis 3.** *Organisational commitment (affective, calculative, and normative) has a negative and significant effect on turnover intentions.*

2.4.3. Mediating Effect of Organisational Commitment on the Relationship between Leadership (Transformational and Transactional) and Turnover Intentions

Effective leadership makes employees feel more committed to the organisation where they work (Fiaz et al. 2017). A higher organisational commitment causes their turnover intentions to decrease (Moreira et al. 2022). This reasoning leads us to the indirect effect of organisational commitment on the relationship between leadership style and turnover intentions. Due to this conceptualisation, we aim to analyse the relationship between leadership (transformational and transactional) and turnover intentions through organisational commitment. This reasoning leads us to the formulation of the following hypothesis:

**Hypothesis 4.** *Organisational commitment mediates the relationship between leadership style (transformational and transactional) and turnover intentions.*

*2.5. Moderating Effect of the Department Where the Employee Works*

Since hospital organisations have many departments, some of which require greater specialisation, the relationship between leadership style and organisational outcome intentions is expected to vary depending on the department of activity. In the field of hospital organisation, it is necessary to guide models that aim to decentralise decision-making and direct it to where care is delivered. Teams are essential in improving performance, and it is important to emphasise strong leadership, recognised by colleagues, which can mobilise professionals for the procedures and actions necessary to make wise use of resources that ensure the organisation's sustainability. Workplace dissatisfaction is considered one of the main causes of turnover among healthcare professionals (Hall et al. 2010). The following hypotheses were developed.

**Hypothesis 5.** *The department to which the employee belongs has a moderating effect on the relationship between leadership style (transformational and transactional) and turnover intentions.*

A theoretical model that synthesizes the formulated hypotheses was developed (Figure 1).

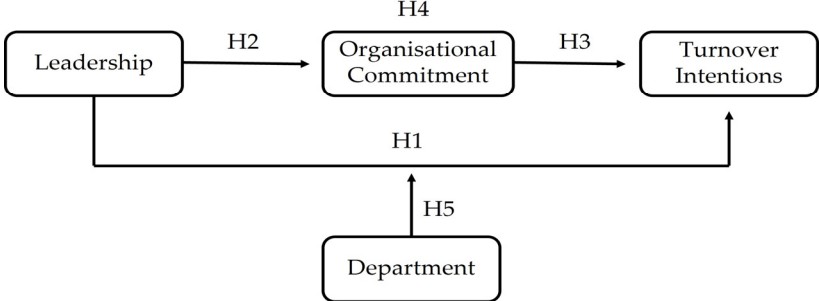

**Figure 1.** Research model.

## 3. Method

### 3.1. Procedure

The data collection technique used in this research was the questionnaire. A request for approval was forwarded to the Clinical Research Unit, the Data Protection Officer, and the Ethics Committee of the HFF, which was granted. Data collection was between 1 April and 29 April 2022. Employees participated voluntarily. Through an e-mail sent by the Clinical Research Unit, the participants were invited to participate in this analysis and complete an online questionnaire on Leadership Styles and Leaving Intentions: Mediation by Affective Organisational Commitment and Moderation by the Activity Department, whose link was provided in the e-mail. The questionnaire was placed online on the Google Docs platform and contained information about the purpose of the analysis. The confidentiality of the answers is also guaranteed since the analysis to be carried out will be of all employees. Employees were also asked for their sincerity, as there were no right or wrong answers since we are only interested in their opinion. Informed consent was presented on the home page of the questionnaire itself and obtained prior to its completion. Participants could start filling out the questionnaire after explicitly indicating that they agreed to participate in the analysis. To this end, a question to this effect was placed on the questionnaire, which should be answered in the affirmative. After informed consent was given, the participants were asked to complete the questionnaire. The questionnaire comprised eight questions to characterise the sample (age, gender, academic qualifications, seniority in the organisation, employment relationship, and department where they work) and three scales (leadership, organisational commitment, and turnover intentions).

### 3.2. Characterization of the Organisation

The HFF was created by Decree-Law no. 382/91 of October 9 and was the first private management experience of an NHS Hospital, having been transformed into a Public Enterprise Entity by Decree-Law 203/2008 of October 10. It is an integrated Hospital in the National Health Service network, and its area of influence is the municipalities of Amadora and Sintra, serving a population of around 550 thousand inhabitants.

This working model focuses on professional competencies and is suitable for organisations that need specific behaviours, talents, and skills to get the job finished effectively (Unger et al. 2009). A professional bureaucracy characterises the HFF; that is, it depends on the knowledge of its employees to carry out its activities and offer services, constituting incremental and emergent decision-making. In this professional bureaucracy model, it is essential to know the team and its competencies to find employees who best fit the crucial functions of the organisation, making their skills establish an interesting competitive advantage (Lunenburg 2012).

As of 31 December 2021, the HFF had 3227 employees working, of which 2522 were women, and 705 were men, thus evidencing a predominance of the female gender (78.2%). The Hospital integrates 49 professionals with disabilities (Table 1).

**Table 1.** Distribution of employees by gender and professional group.

| Professional Group | M | F | Total |
|---|---|---|---|
| Social Organs | 4 | 3 | 7 |
| Directors | 8 | 15 | 23 |
| Physicians | 141 | 287 | 428 |
| Intern Doctors | 92 | 165 | 257 |
| Nurses | 200 | 855 | 1055 |
| Higher Health Technicians | 6 | 34 | 40 |
| Diagnostic and Therapeutic Technicians | 47 | 210 | 257 |
| Higher Technicians | 15 | 49 | 64 |
| Informatics | 7 | 3 | 10 |
| Childhood Educators | | 2 | 2 |
| Technical Assistants | 52 | 241 | 293 |
| Operational Assistants | 133 | 658 | 791 |
| Total | 705 | 2522 | 3227 |

### 3.3. Participants

The sample is composed of 477 participants working at the Hospital Professor Doutor Fernando Fonseca, performing their functions in several departments. Participants in this study are mostly female (69.4%), aged between 19 and 69 years (Mean (M) = 40.41; Standard Deviation (SD) = 10.28) and with seniority in the organisation between 25 and 27 years (M = 12.10; SD = 8.15). As for academic qualifications, 115 (24.1%) have qualifications up to the 12th grade, 216 (45.5%) have a bachelor's degree, and 146 (30.6%) have a master's degree or higher. As for the type of work contract, 79 (16.6%) have a permanent contract, 49 (10.3%) have a fixed-term contract, and 349 (73.2%) have an open-ended contract. Among these participants, 35 (7.3%) work in the children and youth department, 31 (6.5%) in the women's department, 87 (18.2%) in the surgery and specialities department, 85 (17.8%) in the complementary diagnostic and therapeutic means (CDM) department, 125 (26.2%) in the medicine, medical specialities, and emergency department, 36 (7.5%) in the mental health department, 34 (7.1%) in the care support service, 41 (8.6%) in the general support services and 3 (6%) in the technical support services. As for the professional category, 20 (4.2%) are managers, 109 (22.9%) are physicians, 135 (28.3%) are nurses, 33 (6.9%) are diagnostic, and therapeutic technicians, 28 (5.9%) are senior health technicians, 30 (6.3%) are senior technicians, 61 (12.8%) are administrative assistants, and 61 (12.8%) are operational assistants. Among these participants, 200 (41.9%) work in shifts.

### 3.4. Data Analysis Procedure

After data collection, data were imported and analysed into IBM SPSS Statistics version 28 software (IBM Corp., Armonk, NY, USA).

Initially, the metric qualities of the four instruments used in this analysis were tested. Confirmatory factor analyses were performed in the software AMOS 28 for Windows (IBM Corp., Armonk, NY, USA) to test their validity. The program followed the logic of "model generation" (Jöreskog and Sörbom 1993), considering the results obtained interactively in the adjusted analysis: for chi-square ($\chi^2$) $\leq$ 5; for Tucker Lewis Index (NFI) > 0.90; for Fit Goodness Index (GFI) > 0.090; for Comparative Fit Index (CFI) > 0.90; for Root Mean Square Error or Approximation (RMSEA) $\leq$ 0.08; and for the root mean square residual (RMSR), a better adjustment results from a smaller value. Next, the internal consistency of each instrument was checked by calculating Cronbach's alpha, whose value should range between "0" and "1". Negative values were not assumed (Hill and Hill 2002), and greater than 0.70 was an adequate value to organise the research (Bryman and Cramer 2003). For sensitivity analysis, calculations are used for measures of central tendency such as median, skewness, and kurtosis, along with minimum and maximum values for each item.

Hypotheses 1, 2, 3 and 4 were tested using multiple linear regressions. Hypothesis 5, which assumes a moderating effect, was tested using the Macro PROCESS 4.0 (Hayes, New York, NY, USA), developed by Hayes (2013).

*3.5. Instruments*

The Multifactor Leadership Questionnaire (MQL), developed by Bass (1985), was used to measure the leadership style in its version adapted and reduced for Portugal by Salanova et al. (2011). This instrument comprises 28 items and is divided into two subscales: transformational and transactional leadership. The 28 items that make up this instrument are classified on a five-point Likert-type rating scale (from 1 "Never" to 5 "Often if not always"). To test validity, a five-dimensional AFC was performed. Although we had acceptable adjustment indexes ($\chi 2/gL = 3.47$; CFI = 0.97; GFI = 0.90; TLI = 0.97; RMSR = 0.22; RMSEA = 0.072), the five dimensions were highly correlated, with values higher than 0.90. For this reason, a new one-dimensional AFC was performed. As can be seen, the value of the fit indices is similar ($\chi 2/gL = 3.27$; CFI = 0.97; GFI = 0.91; TLI = 0.97; RMSR = 0.019; RMSEA = 0.069), but slightly better for the one-factor model. The transformational leadership subscale consists of 20 items distributed across five dimensions with four items: idealized attributes, idealized behaviours, inspirational motivation, intellectual stimulation, and individualized consideration. For the transactional leadership subscale, a two-dimensional AFC was performed. As the adjustment indexes did not prove to be adequate ($\chi 2/gL = 8.34$; CFI = 0.96; GFI = 0.94; TLI = 0.93; RMSR = 0.072; RMSEA = 0.124) and some items were strongly correlated with other items outside the dimension to which they belong, a new one-factor AFC was performed. As can be seen the adjustment indices are adequate ($\chi 2/gL = 1.21$; CFI = 0.99; GFI = 0.99; TLI = 0.99; RMSR = 0.021; RMSEA = 0.010). As for internal consistency, the transformational leadership subscale has a Cronbach's alpha of 0.98, and the transactional leadership subscale has a value of 0.90. As for construct reliability, transformational leadership has a value of 0.79 and transactional leadership has a value of 0.70. Regarding convergent validity, transactional leadership had an AVE value of 0.77, and transactional leadership had a value of 0.52.

We used the instrument developed by Bozeman and Perrewé (2001) to measure turnover intentions. This instrument is composed of three items, classified on a five-point rating scale (from 1, "I strongly disagree," to 5 ", I strongly agree"). The validity of the exit intention scale was tested by employing exploratory factor analysis since the scale is composed of only three items. A KMO value of 0.73 was obtained, which can be considered reasonable (Sharma 1996), and Bartlett's test of sphericity was significant at $p < 0.001$. It was found that this scale is unidimensional and that this factor explains 86.72% of the total variability of the scale. As for internal consistency, it presents a Cronbach's alpha of 0.92.

To assess the organisational commitment, the scale of Meyer and Allen (1997), in its version adapted for Portugal by Nascimento et al. (2008), was applied. This scale consists of 19 items, divided into three dimensions: affective commitment, with six items (2, 6, 7, 9, 11, and 15); calculative commitment, with seven items (1, 3, 13, 14, 16, 17, and 19); and normative commitment, with six items (4, 5, 8, 10, 12, and 18). Four items are negatively formulated (2, 5, 7, and 15), making it necessary to invert them when rating them. It is a Likert-type scale, with each item being rated on a scale ranging from 1 ("strongly disagree") to 7 ("strongly agree"). To test the validity of this scale, a three-factor AFC was performed. After the AFC was performed, it was found that not all adjustment indices were adequate ($\chi 2/gl = 6.73$; CFI = 0.89; GFI = 0.86; TLI = 0.84; RMSR = 0.056; RMSEA = 0.109) and that some items had factor weights below 0.50. Withdrawing items 5, 9, 14, and 15 for low factor weights or for being highly correlated with dimensions to which they did not belong, a new AFC was performed. This time the adjustment indexes are adequate ($\chi 2/gL = 4.30$; CFI = 0.95; GFI = 0.93; TLI = 0.93; RMSR = 0.039; RMSEA = 0.083). Concerning internal consistency, affective commitment has a Cronbach's alpha value of 0.86, calculative commitment a value of 0.91, and normative commitment a value of 0.91. Regarding construct reliability, the affective commitment had a value

of 0.69, calculative commitment a value of 0.70, and normative commitment a value of 0.79. Concerning convergent validity, the affective commitment had an AVE of 0.52, computational commitment a value of 0.53, and normative commitment a value of 0.66.

Regarding the sensitivity of the items, no item has a median close to one of the extremes, all items have responses at all points, and their absolute values of skewness and kurtosis are below 3 and 7, respectively, so it can be said that they do not grossly violate normality.

### 4. Results

To understand the position of the answers given by the participants to the instruments used in this study, descriptive statistics of the variables under study were performed.

As seen in Table 2, regarding leadership style, the participants in this study perceived their leaders to have a transformational and transactional leadership style, significantly below the midpoint of this scale. Concerning their intentions to leave the organisation, they were revealed to have turnover intentions significantly above the midpoint of the scale (3). Concerning organisational commitment, both affective and normative commitment is below the scale's midpoint (4). However, it should be noted that only the difference concerning normative commitment is significant. Finally, the calculative commitment is above the scale's midpoint (4), although the difference is insignificant.

**Table 2.** Descriptive Statistics of the Variables under Study.

| Variable | t | *p* | Mean | SD |
|---|---|---|---|---|
| Transformational Leadership | −3.02 ** | 0.001 | 2.87 | 0.94 |
| Transational Leadership | −5.11 *** | <0.001 | 2.82 | 0.76 |
| Turnover Intentions | 6.46 *** | <0.001 | 3.35 | 1.25 |
| Affective Commitment | −0.70 | 0.241 | 3.95 | 1.54 |
| Calculative Commitment | 1.53 | 0.063 | 4.11 | 1.63 |
| Normative Commitment | −7.05 *** | <0.0001 | 3.47 | 1.64 |

Note. ** $p < 0.01$; *** $p < 0.001$.

These results indicate that the participants in this study have a low perception of their leader's leadership style, have low levels of affective and normative commitment, high levels of calculative commitment, and have high turnover intentions.

We then tried to understand how these variables varied according to the professional category of the participants (Figure 2).

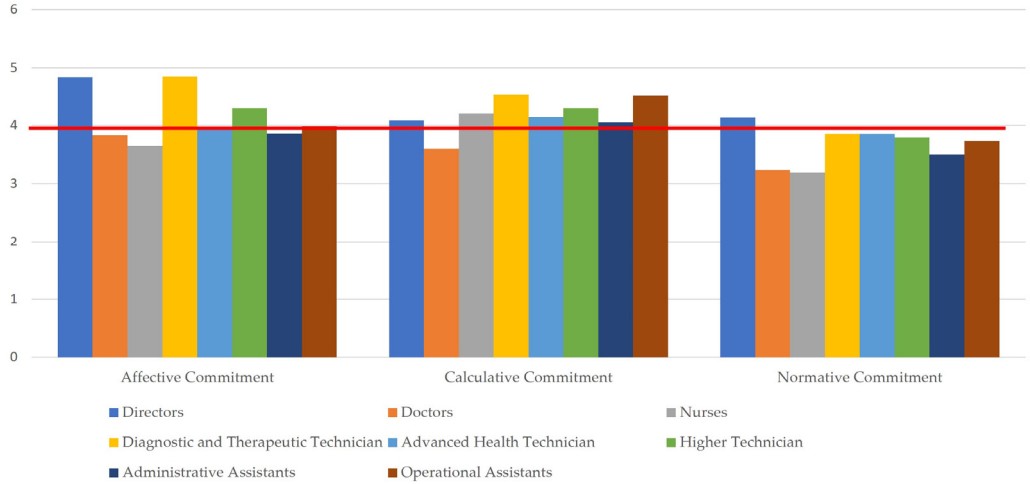

**Figure 2.** Organisational commitment according to professional category.

As shown in Figure 2, nurses (M = 3.64; SD = 1.70) have a lower level of affective commitment and diagnostic and therapeutic technicians (M = 3.64; SD = 1.70) have higher levels. As for normative commitment, medical doctors (M = 3.23; SD = 1.77) and nurses (M = 3.19; SD = 1.64) have the lowest levels and managers (M = 4.13; SD = 1.49) have the highest levels. Finally, regarding calculative commitment, operational assistants (M = 4.52; SD = 1.24) had the highest levels, and physicians (M = 3.60; SD = 1.89) had the lowest levels.

Concerning leadership, nurses (M = 2.64; SD = 0.85) perceived their leaders as having a lower level of transformational leadership style and diagnostic and therapeutic technicians (M = 3.29; SD = 1.02) at a higher level (Figure 3). Regarding transactional leadership, it is the senior health technicians (M = 2.51; SD = 0.65) who perceive their leaders to have a low leadership level, and the diagnostic and therapeutic technicians (M = 3.19; SD = 0.80) have a higher level (Figure 3). Finally, it is the diagnostic and therapeutic technicians (M = 2.61; SD = 1.24) who have the lowest intentions to leave the organisation and the medical doctors (M = 3.41; SD = 1.29) and nurses (M = 3.60; SD = 1.21) the highest (Figure 3).

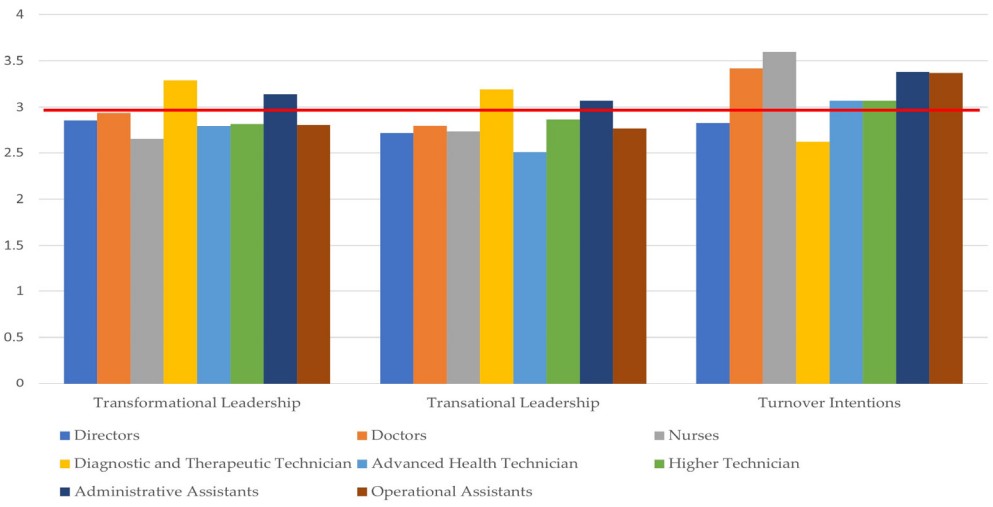

**Figure 3.** Leadership and Turnover Intentions according to professional Category.

Finally, the variability of the variables under study was tested according to the department where the employee works.

It is the employees working in the women's department (M = 2.95; SD = 1.64) who were found to have the lowest levels of affective commitment and those working in general support services (M = 4.88; SD = 1.12) who had the highest levels (Figure 4). As for normative commitment, it is the employees working in the mental health department (M = 2.71; SD = 1.92) who have the lowest levels and those working in technical support services (M = 4.08; SD = 1.91) who have the highest levels (Figure 4). Regarding the calculative commitment, it is the employees working in the child and youth department (M = 3.50; SD = 1.34) who have the lowest levels and those working in the MCDT department (M = 4.55; SD = 1.20) who have the highest levels (Figure 4).

Concerning leadership, it is employees working in the women's department who perceived their leaders as having a lower transformational (M = 2.30; SD = 0.69) and transactional (M = 2.53; SD = 0.65) leadership style and who revealed more turnover intentions (M = 3.94; SD = 0.67) (Figure 5). In turn, it is the employees working in general support services (M = 3.19; SD = 0.80) who perceived their leaders as having a transformational (M = 3.49; SD = 0.86) and transactional (M = 3.28; SD = 0.68) leadership style the highest, and who revealed little turnover intentions (M = 2.52; SD = 1.12) (Figure 5).

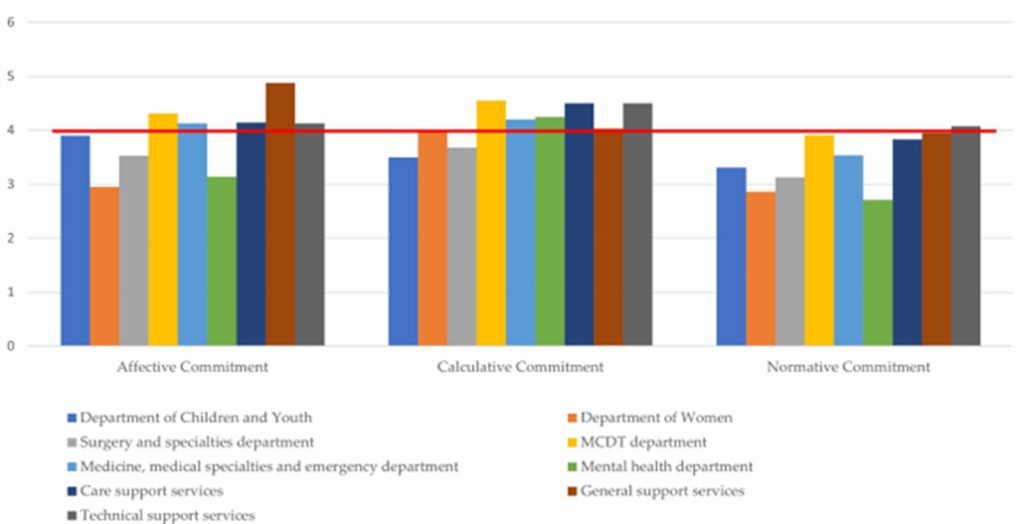

**Figure 4.** Organisational commitment according to department.

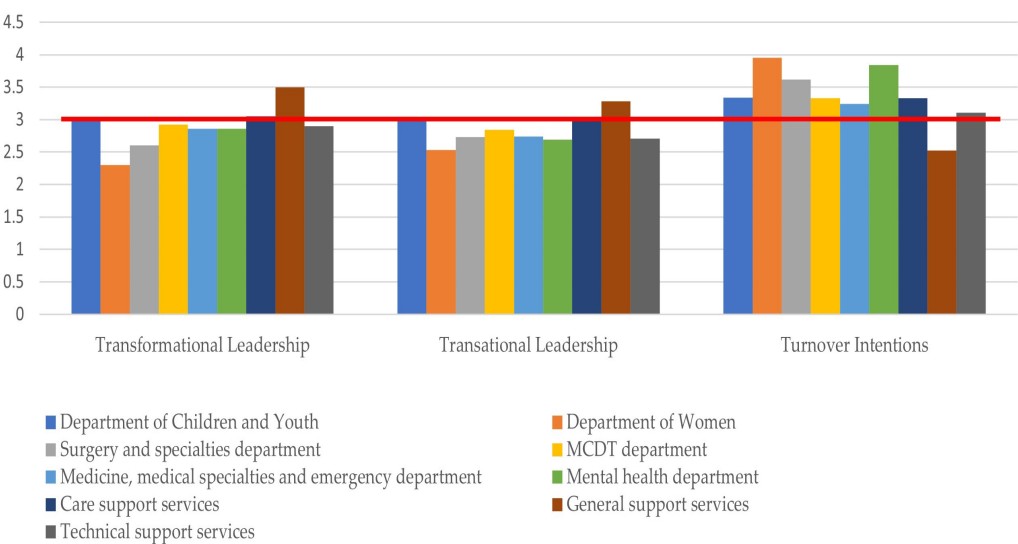

**Figure 5.** Leadership and Exit Intentions by department.

The next step was to study the association between the variables using Pearson's correlations.

Transformational leadership is positively and significantly associated with affective commitment (r = 0.55; $p < 0.001$) and normative commitment (r = 0.37; $p < 0.001$) and negatively and significantly associated with calculative commitment (r = −0.15; $p < 0.001$) (Table 3). Transactional leadership is positively and significantly associated with affective commitment (r = 0.38; $p < 0.001$) and normative commitment (r = 0.24; $p < 0.001$) and negatively and significantly associated with calculative commitment (r = −0.25; $p < 0.001$) (Table 3).

**Table 3.** Association between the variables under study.

|  | 1.1 | 1.2 | 2.1 | 2.2 | 2.3 | 3 |
|---|---|---|---|---|---|---|
| 1.1. Transformational Leadership | – | | | | | |
| 1.2. Transational Leadership | 0.81 *** | – | | | | |
| 2.1. Affective Commitment | 0.55 *** | 0.38 *** | – | | | |
| 2.2. Calculative Commitment | −0.15 ** | −0.25 *** | −0.08 | – | | |
| 2.3. Normative Commitment | 0.37 *** | 0.24 *** | 0.49 *** | 0.12 ** | – | |
| 3. Turnover Intentions | −0.63 *** | −0.50 *** | −0.70 *** | 0.12 * | −0.50 *** | – |

Note. * $p < 0.05$; ** $p < 0.01$; *** $p < 0.001$.

Organisational exit intentions are negatively and significantly associated with transformational leadership ($r = -0.63$; $p < 0.001$), transactional leadership ($r = -0.50$; $p < 0.001$), affective commitment ($r = -0.70$; $p < 0.001$), normative commitment ($r = -0.50$; $p < 0.001$), and positively and significantly associated with calculative commitment ($r = 0.12$; $p = 0.012$) (Table 3).

Finally, the hypotheses formulated in this study were tested. Hypothesis 1 was tested using multiple linear regression (Table 4).

**Table 4.** Leadership and Turnover Intentions (H1a and H1b).

| Independent Variable | Dependent Variable | F | $p$ | $R^2a$ | β | $p$ |
|---|---|---|---|---|---|---|
| Transformational Leaderdhip | Turnover Intentions | 157.69 *** | <0.001 | 0.40 | −0.66 *** | <0.001 |
| Transacional Leadership | | | | | 0.03 | 0.483 |

Note. *** $p < 0.001$.

The results showed only transformational leadership negatively and significantly affected turnover intentions (β = −0.66; $p < 0.001$). The model explained 40% of the variability in turnover intentions. The model is statistically significant (F (2, 474) = 157.69; $p < 0.001$) (Table 4).

Hypothesis 1a is supported, but Hypothesis 1b is not.

To test hypotheses 2a, 2b, 2c, 2d, 2e and 2f, several multiple linear regressions were performed (Table 5).

**Table 5.** Leadership and Organisational Commitment (H2a, H2b, H2c, H2d, H2e and H2f).

| Independent Variable | Dependent Variable | F | $p$ | $R^2a$ | β | $p$ |
|---|---|---|---|---|---|---|
| Transformational Leaderdhip | Affective Commitment | 112.33 *** | <0.001 | 0.32 | 0.73 *** | <0.001 |
| Transacional Leadership | | | | | −0.22 *** | <0.001 |
| Transformational Leaderdhip | Calculative Commitment | 17.41 *** | <0.001 | 0.06 | −0.15 | 0.052 |
| Transacional Leadership | | | | | −0.37 *** | <0.001 |
| Transformational Leaderdhip | Normative Commitment | 39.65 *** | <0.001 | 0.14 | 0.50 *** | <0.001 |
| Transacional Leadership | | | | | −0.16 * | 0.025 |

Nota. * $p < 0.05$; *** $p < 0.001$.

Transformational leadership (β = 0.73; $p < 0.001$) has a positive and significant effect on affective commitment. Transactional leadership (β = −0.22; $p < 0.001$) has a negative and significant effect on affective commitment. The model explains 32% of the variability in affective commitment. The model is statistically significant (F (2, 474) = 112.33; $p < 0.001$) (Table 5).

Hypotheses 2a and 2b were supported.

Only transactional leadership (β = −0.37; $p < 0.001$) negatively and significantly affects calculative commitment. The model explains 6% of the variability in calculative commitment. The model is statistically significant (F (2, 474) = 17.41; $p < 0.001$) (Table 5).

Only Hypothesis 2d was supported.

Transformational leadership (β = 0.50; $p < 0.001$) has a positive and significant effect on normative commitment. Transactional leadership (β = −0.16; $p = 0.025$) has a negative and significant effect on normative commitment. The model explains 14% of the variability in normative commitment. The model is statistically significant (F (2, 474) = 39.65; $p < 0.001$) (Table 5).

Hypotheses 2e and 2f were supported.

To test hypothesis 3, a multiple linear regression was performed (Table 6).

**Table 6.** Organisational Commitment and Turnover Intentions (H3).

| Independent Variable | Dependent Variable | F | *p* | $R^2a$ | β | *p* |
|---|---|---|---|---|---|---|
| Affective Commitment | | | | | −0.58 *** | <0.001 |
| Calculative Commitment | Turnover Intentions | 176.21 *** | <0.001 | 0.53 | 0.10 ** | 0.003 |
| Normative Commitment | | | | | −0.23 *** | <0.001 |

Note. ** *p* < 0.01; *** *p* < 0.001.

Affective commitment (β = −0.58; *p* < 0.001) and normative commitment (β = −0.23; *p* < 0.001) have a significant and negative effect turnover intentions. Calculative commitment (β = 0.10; *p* = 0.003) significantly and positively affects turnover intentions. The model explains 53% of the variability in turnover intentions. The model is statistically significant (F (3, 473) = 176.21; *p* < 0.001) (Table 6).

This hypothesis was supported.

To examine the mediating role of organisational commitment variables between leadership style and willingness to leave the organisation, we followed Baron and Kenny's (1986) procedure, which suggested checking the effect of the three preceding conditions on performance in the mediation test, only meeting the conditions to test the mediating effect of affective and normative commitment on the relationship between transformational leadership and intentions to leave the organisation.

To test this hypothesis, a two-step multiple linear regression was performed. In the first step, the predictor variable was introduced as an independent variable, and in the second step, the mediating variables (Table 7).

**Table 7.** Mediating Effect of Organisational Commitment (H4).

| Independent Variables | Turnover Intentions | |
|---|---|---|
| | β Step 1 | β Step 2 |
| Transformational Leadership | −0.63 *** | −0.33 *** |
| Affective Commitment | | −0.43 *** |
| Normative Commitment | | −0.17 *** |
| *Overall* F | 315.66 *** | 230.95 *** |
| $R^2a$ | 0.40 | 0.59 |
| Δ | | 0.19 *** |

Note. *** *p* < 0.001.

After performing the multiple linear regression test, it is found that by introducing the mediating variables into the regression equation, they have a negative and significant effect on turnover intentions, and the effect of transformational leadership on turnover intentions remains significant but decreases in intensity: M1 (β = −0.63; *p* < 0.001); M2 (β = −0.33; *p* < 0.001) (Table 7). The increase in variability (Δ$R^2a$ = 0.19; *p* < 0.001) is significant. Both models are statistically significant.

In view of these results, it can be stated that affective commitment and calculative commitment partially mediate the relationship between transformational leadership and turnover intentions.

To test Hypothesis 5, we used Macro Process 4.0, developed by Hayes (2013) (Table 8).

It was found that the department does not have a moderating effect on the relationship between leadership (transformational and transactional) and intentions to leave the organisation.

**Table 8.** Moderating effect of the activity department.

| Variable | B | SE | t | p | 95% CI |
|---|---|---|---|---|---|
| Transformational Leadership $\rightarrow$ Turnover Intentions ($R^2 = 0.40$; $p < 0.001$) | | | | | |
| Constant | 5.96 *** | 0.39 | 15.44 *** | <0.001 | [5.20, 6.71] |
| Tranformational Leadership | −0.86 *** | 0.13 | −6.67 *** | <0.001 | [−1.11, −0.60] |
| Department | −0.05 | 0.08 | −0.57 | 0.571 | [−0.21, 0.11] |
| Transformational Leadership * Department | 0.01 | 0.03 | 0.82 | 0.821 | [−0.05, 0.06] |
| Transational Leadership $\rightarrow$ Turnover Intentions ($R^2 = 0.26$; $p < 0.001$) | | | | | |
| Constant | 5.67 *** | 0.52 *** | 10.94 | <0.001 | [4.65, 6.69] |
| Transational Leadership | −0.71 *** | 0.17 *** | −4.08 | <0.001 | [−1.06, −0.37] |
| Department | −0.01 | 0.11 | −0.03 | 0.974 | [−0.22, 0.21] |
| Transational leadership * Department | −0.02 | 0.04 | −0.60 | 0.551 | [−0.09, 0.05] |

Note. *** $p < 0.001$.

## 5. Discussion

This research aimed to study the effect of leadership (transformational or transactional) on organisational exit intentions and whether this relationship is mediated by affective organisational commitment and moderated by the department of activity.

The descriptive statistics of the variables under study showed that the organisational commitment component with the highest mean is the continuity commitment, i.e., employees begin to weigh the pros and cons of leaving the organisation. Among the employees with the highest calculative commitment levels are operational assistants and diagnostic and therapeutic technicians. This is followed by affective commitment, slightly below the center point, which should not be the case in a healthcare organisation. The professionals with the lowest level of affective commitment are nurses and medical doctors, precisely the professional classes most directly linked to patients. Normative commitment is significantly below the central point, and medical doctors and nurses are, again, the professionals with the lowest levels. These results are worrisome. As for the department where the employee works, we found that the professionals who showed the lowest levels of affective and normative commitment are those in the women's department.

As for leadership, both transformational and transactional leadership are significantly below the scale's midpoint, which indicates that it does not exist. The lowest perception of transformational leadership corresponds to nurses and caregivers. Health technicians also have the lowest perception of transactional leadership. In turn, the professionals in the women's department have the lowest perception of transformational and transactional leadership.

Finally, turnover intentions are significantly above the central point, indicating that the participants in this study have high intentions to leave the organisation soon. The professionals with the highest turnover intentions are medical doctors and nurses. Concerning the department, they are professionals working in the women's department.

First, transformational leadership's negative and significant effect on turnover intentions was proven. These results align with what the literature tells us since, according to Park and Pierce (2020), transformational leadership has a negative and significant effect on turnover intentions. When a leader encourages his or her employees to play a critical role in decision making, to perform their jobs with a focus beyond the short-term goals through influence, to raise maturity levels and ideals, as well as to encourage self-fulfillment concerns in each employee (Bass 1999), it makes them want to stay in the organisation where they work. This relationship can be interpreted based on the Resource Conservation Theory since employees perceive their leader's leadership style as transformational and want to stay with the organisation (Hobfoll 1989; Pinto and Chambel 2008). The positive and significant effect of transactional leadership on turnover intentions was not proven, which goes against what some authors report since, for Koesmono (2017), transactional leadership

has a positive and significant effect on turnover intentions. However, the results of this study are in line with the results obtained by Islam et al. (2012), who also did not find a significant relationship between transactional leadership and turnover intentions. These results may be because there are rewards and recognition for employees from their bosses. The data from this study was collected after a pandemic, and the entire society recognized the efforts of the healthcare workers.

Secondly, transformational leadership was found to have a positive and significant effect on affective and normative commitment, but its effect on calculative commitment was not significant. These results align with the study by Zaraket and Sawma (2018), in which transformational leadership only has a positive and significant effect on affective and normative commitment, while its effect on calculative commitment was not found to be significant. A transformational leader stimulates and motivates his or her employees, which makes them feel affectively and morally attached to the organisation. When the employee perceives his leader's leadership style as transformational, his way of rewarding it will be to stay in the organisation. The relationship can be interpreted based on the premise of social exchanges (Blau 1964) and the norm of reciprocity (Gouldner 1960). There was also evidence of a significant negative effect of transactional leadership on all three components of organisational commitment. These results are also in line with the study by Ngunia et al. (2006) whereby transactional leadership had a significant and negative effect on organisational commitment.

Thirdly, the negative and significant effect of affective and normative commitment on turnover intentions was confirmed. These results align with what several authors have reported, including Meyer et al. (2002), that organisational commitment is a reducer of turnover intentions. However, the calculative commitment had a positive and significant effect on turnover intentions, which goes against what the literature tells us. Possibly, employees weighed the pros and cons of leaving the organisation where they work and thought that they would benefit from leaving the organisation, which is now a reality since there are many offers to work in the private sector, sometimes with better conditions, both monetarily and in terms of career progression. Among the three components of organisational commitment, the strongest effect on turnover intentions was affective commitment, again confirming the results found by other authors such as Moreira and Cesário (2021). Some authors, including Meyer and Allen (1991), state that the great interest in studying affective commitment is because it is the best reducer of turnover intentions.

Fourthly, the partial mediating effect of affective and normative commitment on the relationship between transformational leadership and turnover intentions was proven. When a leader encourages his or her employees to play a critical role in decision making, to perform his or her duties with a focus beyond short-term objectives through the influence he or she exerts, managing to increase maturity levels, ideals, as well as stimulating the existence of concerns with each employee's self-fulfillment (Bass 1999), he or she is enhancing their affective and normative commitment, reducing their turnover intentions.

Finally, the moderating effect of the department to which the employee belongs on the relationship between leadership and turnover intentions was not proven. These results may be because the leadership averages (transformational and transactional) only depend a little on the department where the employee works.

The results of this study are worrisome since the levels of leadership (transformational and transactional) and organisational commitment (affective and normative) are very low, as opposed to the levels of calculative commitment and turnover intentions, which are high. These results reflect a need for nurses and medical doctors in public hospitals since their departure to the private sector has been accentuated. The same happens to professionals in the women's sector, which reflects the number of days that obstetrics emergency rooms have been closed recently.

The leaders of the HFF should have more transformational leadership to foster organisational commitment in their subordinates, thus reducing turnover intentions.

*Limitations*

This study has some limitations. The main limitation is that it is a cross-sectional study, which did not allow for establishing causal relationships between the variables. It would be necessary to conduct a longitudinal study, to test causal relationships. Another limitation is that self-report questionnaires were used, which may have biased the results. Several methodological and statistical recommendations were followed to reduce the impact of common method variance (Podsakoff et al. 2003).

Finally, one limitation is that this study was conducted after a pandemic, and no exit interviews or questionnaires were answered during this period. The same study should be replicated in a year or two to consistent results.

## 6. Conclusions

Healthcare organisations provide complex and highly specialized services and come into contact with vulnerable and dependent users. According to Bolton et al. (2021), creating an organisational culture that supports continuous improvement supports the necessary connection between professionals around the patient. For Crossan et al. (1999), organisational learning is a process of change in individual thinking and action, affected by and inscribed in the organisation's institutions. For Morais (2012, p. 29), leadership in healthcare organisations affects how the service is delivered, how its goals are achieved, and performance." The abandonment of employees in the health sector is a critical challenge worldwide. In this sense, the turnover intention is an intervening variable between actual leaving and job satisfaction and is therefore affected by similar individual and organisational factors.

As a strength of this study, we have the fact that affective and normative commitment are the mechanisms that explain the relationship between transformational leadership and organisational turnover intentions. Organisations should focus on retaining their best employees, as they are hard-to-imitate resources and, according to the theory of the "resource-based perspective" (Afiouni 2007; Barney 1991), become their competitive advantage in today's labour market. The need to implement retention and job satisfaction policies, particularly in public sector healthcare organisations, has been highlighted. Presumably, many retention policies in the public sector are linked to government policies on contract stability, training, careers, and compensation.

Human resource management is critical to creating and developing a competitive, productive, and motivated workforce (Lepak et al. 2007). At this stage, it is primarily about people and their results, transforming the organisation's potential into real opportunities. In other words, people are needed for the organisation to achieve its goals and fulfil its mission, and good people management is essential for the organisation's adaptation and survival.

An organisational culture based on human relationships should be implemented by valuing and recognizing the health professionals involved, encouraging their satisfaction and motivation while fostering group spirit and teamwork.

Thus, we conclude that organisations should be concerned with fostering leaders who assume a more transformational than transactional leadership.

As implications for the literature, this study confirms the importance of transformational leadership in enhancing employees' affective and normative commitment (Fiaz et al. 2017) and reducing their turnover intentions (Moreira et al. 2022).

**Author Contributions:** Conceptualization, P.M. and G.N.; methodology, P.M. and G.N.; software, A.M.; validation, P.M., G.N. and A.M.; formal analysis, A.M.; investigation, P.M. and G.N.; resources, P.M.; data curation, A.M.; writing—original draft preparation, P.M., G.N. and A.M.; writing—review and editing, P.M. and G.N.; visualization, P.M. and A.M..; supervision, G.N. and A.M.; project administration, P.M. and G.N.; funding acquisition, A.M. All authors have read and agreed to the published version of the manuscript.

**Funding:** This research received no external funding.

**Institutional Review Board Statement:** Ethical review and approval were waived for this study since all participants before answering the questionnaire had to read the informed consent and agree to it. This was the only way they could answer the questionnaire. Participants were informed about the purpose of the study, as well as that the results were confidential, as individual results would never be known, but would only be analysed in the set of all participants.

**Informed Consent Statement:** Informed consent was obtained from all subjects involved in the study.

**Data Availability Statement:** The data presented in this study are available on request from the corresponding author. The data is not publicly available because in their informed consent, participants were informed that the data was confidential and that individual responses would never be known, as data analysis would be of all participants combined.

**Conflicts of Interest:** The authors declare no conflict of interest.

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
