# Peer review of "Leadership and Turnover Intentions in a Public Hospital: The Mediating Effect of Organisational Commitment and Moderating Effect by Activity Department"

_admsci, doi:10.3390/admsci13010018_

Round 1

Reviewer 1 Report

Thank you for providing an interesting and well-organized study. The study is mainly well framed and presented, the hypotheses are adequately argued, and the analyses of the data is sufficiently clear. The results are also clearly presented. The discussion is also clear but might be developed by integrating parts of the conclusion chapter, which I conceive of as too long.

This brings me to present a few critical comments.

·       First, the term effective leadership should be briefly explained and defined, as it plays a central role in the argumentation.

·       It sounds more appropriate to claim that the hypotheses are ‘supported instead’ of ‘proven’

·       Rework the conclusion part by shortening it and integrating the arguments in the discussion.

·       The implications from the study should be elaborated and discussed in the and presented briefly in the Conclusions

·       There are a few sentences that are hard to understand and some typos to correct, hence a new proofreading should be considered.

Good luck!

Author Response

Article

- Revision 1 -

Dear Reviewer,

We appreciate your preliminary comments that will complement our work.

Comment 1:  First, the term effective leadership should be briefly explained and defined, as it plays a central role in the argumentation.

In the introduction, a definition of leadership and effective leader was inserted (lines 63 to 69).

Comment 2: It sounds more appropriate to claim that the hypotheses are ‘supported instead’ of ‘proven’.

In all cases "proven" was replaced by "supported.

Comment 3:   Rework the conclusion part by shortening it and integrating the arguments in the discussion.

The discussion has been improved to make it more understandable, and the conclusions have been shortened.

Comment 4: The implications from the study should be elaborated and discussed in the and presented briefly in the Conclusions.

The theoretical implications have been added at the end of the conclusions (lines 849 to 851).

Comment 5:   There are a few sentences that are hard to understand and some typos to correct, hence a new proofreading should be considered.

The sentences that we considered not to be correct were reformulated.

Yours sincerely,

On behalf of my co-authors,

References added to the manuscript:

(Blau, 1964) Blau, Peter Michael 1964. Exchange and Power in Social Life. New York: Wiley.

(Gouldner, 1960) Gouldner, Alvin W. 1960. The norm of reciprocity: A preliminary statement. American Sociological Review, 25, 161–178.

(Hobfoll, 1989) Hobfoll, Stevan E. 1989. Conservation of resources: A new attempt at conceptualizing stress. American Psychologist, 44, 513-524.

(Perez and Oliveira (2015) Perez, Olivia Cristina, and Ana Paula Modesto Oliveira. 2015. Liderança eficaz: o poder e a influência de um líder no comportamento organizacional de uma empresa. Administração de empresas em revista1(10), 1-16.

(Pinto and Chambel, 2008) Pinto, Alexandra Marques and Maria José Chambel. 2008. Abordagens teóricas no estudo do burnout e do engagement. In Alexandra Marques Pinto and Maria José Chambel (Orgs.), Burnout e engagement em contexto organizacional: Estudos com amostras portuguesas (pp. 53-84). Lisboa: Livros Horizonte.

References removed from the manuscript:

(Afiouni, 2007) Afiouni, Fida. 2007. Human Resource Management and Knowledge Management: A Road Map Toward Improving Organizational Performance. Journal of American Academy of Business 11: 124–30.

(Barney, 1991) Barney, Jay B. 1991. Firm resources and sustained competitive advantage. Journal of Management 17, 99–120.

Reviewer 2 Report

I would like to congratulate the authors for their effort studying the relationship between leadership and turnover intentions in a sector so important for society as it is the health one.

However, to improve the quality and readability of the paper, this reviewer would like to offer some humble  comments and suggestions:

a)       The sample should be explained in its own section in the paper. We can see its extension only in the abstract, and a characterisation of it is needed (men, women,  anyone has quitted,…).

b)      Sentences in the paragraphs need bridges (conjunctions,…) between them. If not, they seem juxtaposed as if we were reading a telegram.

c)       In the conclusions, one can easily find the ones for managers but not so much for the academic community.

This reviewer thinks that with a little effort form the authors, the paper can become excellent.

Author Response

Article

- Revision 1 -

Dear Reviewer,

We are very thankful for your interesting insights.

Comment 1: The sample should be explained in its own section in the paper. We can see its extension only in the abstract, and a characterisation of it is needed (men, women,  anyone has quitted,…).

The participants' descriptions were added to the manuscript (lines 430 to 448). It was our oversight not to include it.

Comment 2:   Sentences in the paragraphs need bridges (conjunctions,…) between them. If not, they seem juxtaposed as if we were reading a telegram.

The sentences that we considered not to be correct were reformulated.

Comment 3: In the conclusions, one can easily find the ones for managers but not so much for the academic community.

The theoretical implications have been added at the end of the conclusions (lines 849 to 851).

Yours sincerely,

On behalf of my co-authors,

References added to the manuscript:

(Blau, 1964) Blau, Peter Michael 1964. Exchange and Power in Social Life. New York: Wiley.

(Gouldner, 1960) Gouldner, Alvin W. 1960. The norm of reciprocity: A preliminary statement. American Sociological Review, 25, 161–178.

(Hobfoll, 1989) Hobfoll, Stevan E. 1989. Conservation of resources: A new attempt at conceptualizing stress. American Psychologist, 44, 513-524.

(Perez and Oliveira (2015) Perez, Olivia Cristina, and Ana Paula Modesto Oliveira. 2015. Liderança eficaz: o poder e a influência de um líder no comportamento organizacional de uma empresa. Administração de empresas em revista1(10), 1-16.

(Pinto and Chambel, 2008) Pinto, Alexandra Marques and Maria José Chambel. 2008. Abordagens teóricas no estudo do burnout e do engagement. In Alexandra Marques Pinto and Maria José Chambel (Orgs.), Burnout e engagement em contexto organizacional: Estudos com amostras portuguesas (pp. 53-84). Lisboa: Livros Horizonte.

References removed from the manuscript:

(Afiouni, 2007) Afiouni, Fida. 2007. Human Resource Management and Knowledge Management: A Road Map Toward Improving Organizational Performance. Journal of American Academy of Business 11: 124–30.

(Barney, 1991) Barney, Jay B. 1991. Firm resources and sustained competitive advantage. Journal of Management 17, 99–120.

Reviewer 3 Report

Thank you for submitting the manuscript id admsci-2076814 entitled “Leadership and turnover intentions in a public hospital: the mediating effect of organizational commitment and moderating effect by activity department.” A lot of research studies are available on these variables e.g., leadership, commitment, and turnover intention. Thus require a substantial gap to get the interest of the researchers and practitioners.

Good luck

Abstract
In the abstract, This research aimed to study the effect of leadership (transformational or transactional) on turnover intentions and whether this relationship is mediated by organizational commitment and moderated by the department of activity.It should be transformational and transactional. In addition, it is better to include one or two lines regarding the adopted method.

Introduction
The problem needs to be identified and the gap has to be further illuminated. There are many studies available that have examined the effect of leadership on commitment via turnover intention. E.g.,

Ennis, M. C., Gong, T., & Okpozo, A. Z. (2018). Examining the mediating roles of affective and normative commitment in the relationship between transformational leadership practices and turnover intention of government employees. International Journal of Public Administration, 41(3), 203-215.

Gyensare, M. A., Kumedzro, L. E., Sanda, A., & Boso, N. (2017). Linking transformational leadership to turnover intention in the public sector: The influences of engagement, affective commitment and psychological climate. African Journal of Economic and Management Studies.

Ausar, K., Kang, H. J. A., & Kim, J. S. (2016). The effects of authentic leadership and organizational commitment on turnover intention.
Leadership & Organization Development Journal.

Lim, A. J. P., Loo, J. T. K., & Lee, P. H. (2017). The impact of leadership on turnover intention: The mediating role of organizational commitment and job satisfaction.
Journal of Applied Structural Equation Modeling, 1(1), 27-41.

Gul, S., Ahmad, B., Rehman, S. U., Shabir, N., & Razzaq, N. (2012). Leadership styles, turnover intentions and the mediating role of organizational commitment. In Information and Knowledge Management (Vol. 2, No. 7, pp. 44-51).

Gyensare, M. A., Anku-Tsede, O., Sanda, M. A., & Okpoti, C. A. (2016). Transformational leadership and employee turnover intention: The mediating role of affective commitment.
World Journal of Entrepreneurship, Management and Sustainable Development.

Jang, J., & Kandampully, J. (2018). Reducing employee turnover intention through servant leadership in the restaurant context: A mediation study of affective organizational commitment.
International Journal of Hospitality & Tourism Administration, 19(2), 125-141.
e.t.c

So how your study is different from previous studies?

In addition, there is no justification for the moderator e.g., department. A solid reason is required for adding it as a boundary condition.

Further, how your study contributes to the literature? Briefly add to the last paragraph of the introduction.

Literature review
In general, the literature on the study constructs is well presented. However, the arguments leading to the hypothesis are very thin and need to be strengthened. In addition, the anchoring of theory is recommended in the development of hypotheses.

Methods and results
ď‚· Regarding methods, these things need to be elaborated

Could you please justify why only one hospital was targeted?
When the data was collected?
Since the data were cross-sectional, how did you avoid common method bias (CMB) during data collection? Which statistical tests were applied to detect CMB?
There is also a need to justify the use of SPSS.

ď‚· Results are fine

Discussion and implication
The researchers tried their best to explain the results and provide implications. Unfortunately, I could not find many novel implications of this study.

Other comments
The in-text citation is not consistent. Please check the in-text citation for typos. e.g., page 4. (Moriano et al. (2014), (Shore and Martin, 1989), (Tett & Meyer, 1993; Benson, 2006) among others.

The paper needs proofreading as there are some grammatical errors.

Author Response

Article

- Revision 1 -

Dear Reviewer,

We appreciate your preliminary comments that will complement our work.

Comment 1:  

Abstract
In the abstract, “This research aimed to study the effect of leadership (transformational or transactional) on turnover intentions and whether this relationship is mediated by organizational commitment and moderated by the department of activity.” It should be transformational and transactional. In addition, it is better to include one or two lines regarding the adopted method.

The proposed changes have been made.

Comment 2: The problem needs to be identified and the gap has to be further illuminated. There are many studies available that have examined the effect of leadership on commitment via turnover intention.

So how your study is different from previous studies?

This study differs from previous ones in that it was carried out in a public hospital on the outskirts of Lisbon, the capital of Portugal. This hospital is located in one of the country's largest and most diverse population centers. This human and territorial coverage allows for a diverse set of experiences in the hospital environment. In addition, after the COVID-19 pandemic, the hospital sector in Portugal is going through a great crisis, with many professionals (especially doctors and nurses) leaving public hospitals. The characteristics of this hospital have been added in the introduction (lines 86 to 96).

Comment 3:   In addition, there is no justification for the moderator e.g., department. A solid reason is required for adding it as a boundary condition.

The justification for the moderating effect was added in the introduction (lines 91 to 96).

Comment 4: Further, how your study contributes to the literature? Briefly add to the last paragraph of the introduction.

The contribution to the literature was added in the last paragraph of the introduction (lines 101 to 103).

Comment 5:   In general, the literature on the study constructs is well presented. However, the arguments leading to the hypothesis are very thin and need to be strengthened. In addition, the anchoring of theory is recommended in the development of hypotheses.

The arguments leading to the formulation of the hypotheses have been added (lines 229 to 237; lines 304 to 308).

Comment 6: Regarding methods, these things need to be elaborated.

Could you please justify why only one hospital was targeted?

This hospital was chosen because it is going through difficult times with the departure of many employees, mainly doctors and nurses.

Comment 7:   When the data was collected?

The data collection date has been added (lines 382 to 383). By mistake, it had not been provided.

Comment 8:   Since the data were cross-sectional, how did you avoid common method bias (CMB) during data collection? Which statistical tests were applied to detect CMB? There is also a need to justify the use of SPSS.

Different platforms were used for sending the questionnaire, such as e-mail and private messages through LinkedIn and QR codes. The sample is significant, representing about 15% of all hospital employees. The initial sample consisted of 499 participants, but 22 were excluded for not giving informed consent.

SPSS Statistics software was used since it is the software used in the research centers where the authors of this manuscript work, as it is considered the most appropriate software for this type of study.

Comment 9:  

Discussion and implication

The researchers tried their best to explain the results and provide implications. Unfortunately, I could not find many novel implications of this study.

The implications may not be very innovative in general, but they are innovative and a recurring problem for healthcare institutions in Portugal.

Comment 10:   The in-text citation is not consistent. Please check the in-text citation for typos. e.g., page 4. (Moriano et al. (2014), (Shore and Martin, 1989), (Tett & Meyer, 1993; Benson, 2006) among others.

These errors have been corrected throughout the text.

Comment 7:   The paper needs proofreading as there are some grammatical errors.

The sentences that we considered not to be correct were reformulated.

Yours sincerely,

On behalf of my co-authors,

References added to the manuscript:

(Blau, 1964) Blau, Peter Michael 1964. Exchange and Power in Social Life. New York: Wiley.

(Gouldner, 1960) Gouldner, Alvin W. 1960. The norm of reciprocity: A preliminary statement. American Sociological Review, 25, 161–178.

(Hobfoll, 1989) Hobfoll, Stevan E. 1989. Conservation of resources: A new attempt at conceptualizing stress. American Psychologist, 44, 513-524.

(Perez and Oliveira (2015) Perez, Olivia Cristina, and Ana Paula Modesto Oliveira. 2015. Liderança eficaz: o poder e a influência de um líder no comportamento organizacional de uma empresa. Administração de empresas em revista1(10), 1-16.

(Pinto and Chambel, 2008) Pinto, Alexandra Marques and Maria José Chambel. 2008. Abordagens teóricas no estudo do burnout e do engagement. In Alexandra Marques Pinto and Maria José Chambel (Orgs.), Burnout e engagement em contexto organizacional: Estudos com amostras portuguesas (pp. 53-84). Lisboa: Livros Horizonte.

References removed from the manuscript:

(Afiouni, 2007) Afiouni, Fida. 2007. Human Resource Management and Knowledge Management: A Road Map Toward Improving Organizational Performance. Journal of American Academy of Business 11: 124–30.

(Barney, 1991) Barney, Jay B. 1991. Firm resources and sustained competitive advantage. Journal of Management 17, 99–120.

Round 2

Reviewer 3 Report

Thank you for submitting the revised manuscript entitled "Leadership and turnover intentions in a public hospital: the mediating effect of organizational commitment and moderating effect by activity department." I am happy with the revision.

Good luck with your future endeavors